# Pea Aphid (*Acyrthosiphon pisum*) Host Races Reduce Heat-Induced Forisome Dispersion in *Vicia faba* and *Trifolium pratense*

**DOI:** 10.3390/plants12091888

**Published:** 2023-05-06

**Authors:** Maria K. Paulmann, Linus Wegner, Jonathan Gershenzon, Alexandra C. U. Furch, Grit Kunert

**Affiliations:** 1Max Planck Institute for Chemical Ecology, Department of Biochemistry, Hans-Knöll-Str. 8, D-07745 Jena, Germany; 2Plant Physiology, Matthias Schleiden Institute for Genetics, Bioinformatics and Molecular Botany, Faculty of Biological Science, Friedrich Schiller University Jena, Dornburger Straße 159, D-07743 Jena, Germany; 3Institute of Botany, Justus Liebig University, Heinrich-Buff-Ring 38, 35292 Giessen, Germany

**Keywords:** pea aphid host race, aphid saliva, phloem located defense, phloem protein, sieve element occlusion, calcium, legume, *Pisum sativum*, *Trifolium pratense*, *Vicia faba*

## Abstract

Although phloem-feeding insects such as aphids can cause significant damage to plants, relatively little is known about early plant defenses against these insects. As a first line of defense, legumes can stop the phloem mass flow through a conformational change in phloem proteins known as forisomes in response to Ca^2+^ influx. However, specialized phloem-feeding insects might be able to suppress the conformational change of forisomes and thereby prevent sieve element occlusion. To investigate this possibility, we triggered forisome dispersion through application of a local heat stimulus to the leaf tips of pea (*Pisum sativum*), clover (*Trifolium pratense*) and broad bean (*Vicia faba*) plants infested with different pea aphid (*Acyrthosiphon pisum*) host races and monitored forisome responses. Pea aphids were able to suppress forisome dispersion, but this depended on the infesting aphid host race, the plant species, and the age of the plant. Differences in the ability of aphids to suppress forisome dispersion may be explained by differences in the composition and quantity of the aphid saliva injected into the plant. Various mechanisms of how pea aphids might suppress forisome dispersion are discussed.

## 1. Introduction

Translocation of photoassimilates, nitrogen-rich compounds and signaling molecules in vascular plants is carried out by the transport conduits of the phloem—the sieve elements (SEs). However, SEs are targeted by aphids and other piercing–sucking insects that feed on phloem sap [1,2]. Plants have developed two main mechanisms to prevent the loss of phloem sap in response to mechanical damage [3] or phloem-feeding insects [4,5,6]. (1) Long-term occlusion of SEs is achieved by the calcium-ion (Ca^2+^)-triggered accumulation of callose at the sieve pores and the plasmodesmata, while (2) short-term occlusion is achieved by phloem proteins (P-proteins) [3,7,8,9,10,11]. Among the occluding P-proteins are the spindle-like forisomes, which are unique to legumes [11,12]. In response to damage and subsequent Ca^2+^ influx into the SE, forisomes expand reversibly until they are fully dispersed and clog the SE [10,13,14]. Based on this behavior, forisomes can be considered as natural Ca^2+^ indicators [15]. Forisomes have been shown to stop mass flow in vitro [16] and are believed to have a similar function *in planta* [11,17]. The expansion of forisomes occurs at a much faster rate than callose deposition, and so reduces phloem sap loss until callose deposition is complete [3].

Aphids typically cause little damage to plant tissue because they use their stylets to navigate in between plant cells [18,19,20]. During this process, aphids inject saliva into the plant [21,22]. One type of saliva, called gelling saliva, is secreted into the apoplastic space where it gelatinizes, and the aphid subsequently pushes its stylet through the gel so that the gel forms a sheath [23]. Besides protecting the stylet from mechanical damage and possibly manipulating plant responses [21], the gelling saliva has been suggested to seal cell wall sites where non-SE cells have been pierced [19,24]. This could, for example, prevent influx of Ca^2+^ ions and subsequent aphid recognition by the plant. Elicitors present in the aphid saliva can also lead to plant recognition of the aphid’s presence [25]. During navigation through the plant apoplast, the aphid occasionally pierces plant cells to orient itself [19,26] and secretes a second, watery type of saliva into the cell [27]. Elicitors such as Mp10 may then be recognized by the plant [28], which launches plant defense responses that reduce aphid fecundity [28]. Induced defenses, such as callose and forisomes, can be located in the phloem [29,30,31,32] leading to SE or stylet clogging, which impairs the uptake of sap [5,33] and consequently aphid performance. 

Plant defense responses to phloem feeding can in turn be manipulated by effectors in the aphid saliva [26,34,35]. These effectors may prevent hypersensitive cell death or suppress other defenses, such as Ca*^2+^*-triggered occlusion via callose or forisomes [36,37,38,39]. For example, forisome dispersion is reversible in vitro through application of the saliva of the aphid *Megoura viciae* [Buckton] [38]. Inter- and intraspecific variation of aphid saliva composition [37,40,41] could therefore be responsible for preventing phloem defenses such as occlusion due to forisomes. Saliva composition differences could be involved in the ability of the specialist aphid *Acyrthosiphon pisum* [Harris] to feed readily on *Vicia faba* [L.] and not induce forisome dispersion while the generalist aphid *Myzus persicae* [Sulzer] is not able to establish feeding on *V. faba*, a plant on which it triggers forisome dispersion [4,42].

The specialist *A. pisum* is a species complex of at least 15 genetically different host races. Each host race is native to one or a very few closely related legume species and can survive and reproduce successfully only on its respective host plant [43,44]. However, all pea aphid host races are able to feed on *V. faba*, the so-called universal host plant. Plant responses to pea aphid feeding such as changes in phytohormones [45] and metabolites [46] have been shown to depend on the pea aphid host race. These pea aphid host race-specific responses may be triggered through differences in aphid saliva composition [47]. Feeding behavior of different pea aphid host races indicated that the phloem might be involved in this pea aphid—host plant specificity [31]. Thus, forisomes located in SEs probably play a role in this aphid—host plant specificity. Hence, we hypothesize that the pea aphid host races are able to suppress forisome dispersion on their native host plant and are thus able to feed, but are not able to suppress, forisome dispersion on nonhost plants. 

Since forisome dispersion induced by individual aphids is very difficult to detect and investigate in an experimental context [4,42,48], we triggered dispersion with a heat stimulus and then examined whether pea aphid host races were able to prevent forisome dispersion on native host and nonhost plants as well as on the universal host plant. We found that pea aphid host races were able to suppress heat-triggered forisome dispersion. This however, was dependent on the specific plant species—pea aphid host race combination and on plant age. To see whether the pea aphid host races used for this investigation differed in their saliva composition, watery saliva of different pea aphid host races was collected, and salivary proteins were compared. We found that aphid saliva composition differed between the pea aphid host races and might therefore indeed be responsible for the host race-dependent forisome reaction.

## 2. Results

### 2.1. Suppression of Forisome Dispersion Depends on Host Plant—Aphid Host Race Combination as Well as on Plant Age

The ability of different *A. pisum* host races to suppress forisome dispersion in different host plants was studied through application of a heat stimulus to the leaf tip while monitoring forisome responses in the same leaf. Heat stimulation is a strong trigger for forisome dispersion [3]. We assessed the ability of aphids to reduce forisome dispersion after 48 h of aphid infestation.

For all investigated legume species, the distance of the forisome to the heat-stimulus site did not influence the ratio of dispersed to all forisomes (Table 1).

**Table 1 plants-12-01888-t001:** Statistical values for the analyses of the forisome dispersion ratios of different legume species in response to heat stimulation with variation in the distance of the observed forisome to the site of heat stimulation, variation in plant age, and variation in the host race of the infesting pea aphid (*Acyrthosiphon pisum*; treatment).

Plant	Explanatory Variable	Deviance	*p*-Value
***Vicia faba*** (**Figure 1**)			
	distance	−0.777	0.378
	plant age	−0.302	0.582
	treatment	−2.191	0.534
	**treatment: plant age**	−7.858	**0.049**
	distance: plant age	−0.001	0.970
	distance: treatment	−0.857	0.836
	distance: treatment: plant age	−0.777	0.396
***Trifolium pratense*** (**Figure 2**)			
	distance	−0.006	0.939
	**plant age**	−4.643	**0.031**
	**treatment**	−9.589	**0.022**
	treatment: plant age	−2.284	0.516
	distance: plant age	−0.063	0.802
	distance: treatment	−3.486	0.323
	distance: treatment: plant age	−0.453	0.929
***Pisum sativum*** (**Figure 3**)			
	distance	−0.646	0.422
	treatment	−2.720	0.437
	distance: treatment	−1.880	0.598

Significant *p*-values are highlighted in **bold**.

**Figure 1 plants-12-01888-f001:**
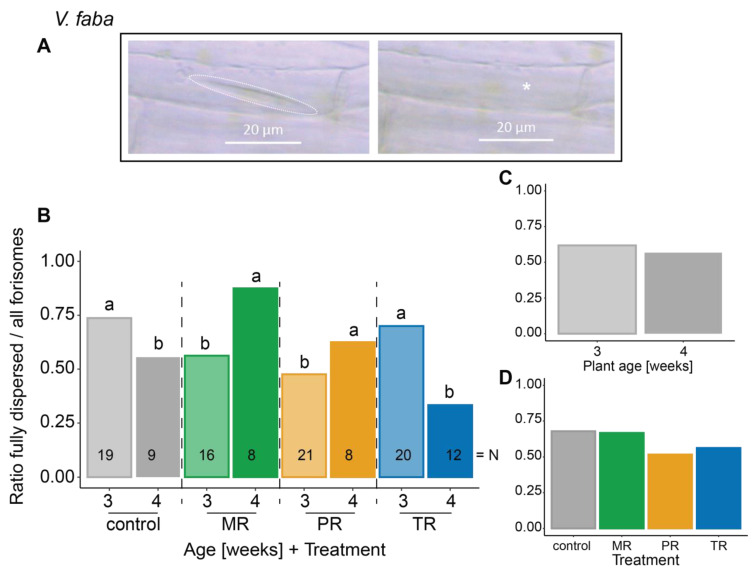
Influence of different *A. pisum* host races on the dispersion behavior of forisomes in the universal host plant *V. faba*. (**A**) *V. faba* forisome in its condensed (left panel, dashed line) and its dispersed (right panel, putative location indicated by asterisk) state. Depicted are the ratios of fully dispersed forisomes to all observed forisomes after heat stimulation (**B**) in various host race treatments on plants at both ages, (**C**) in various plant ages (indicated by fill shade), and (**D**) in various aphid host race treatments. Different lower-case letters above bars indicate significant differences (*p* ≤ 0.05). Data were analyzed by a Bernoulli GLM. Further statistical information can be found in Table 1. Numbers in the bars indicate the total number of forisomes observed for the respective treatment (N). MR—*Medicago* host race (green); PR—*Pisum* host race (orange); TR—*Trifolium* host race (blue).

**Figure 2 plants-12-01888-f002:**
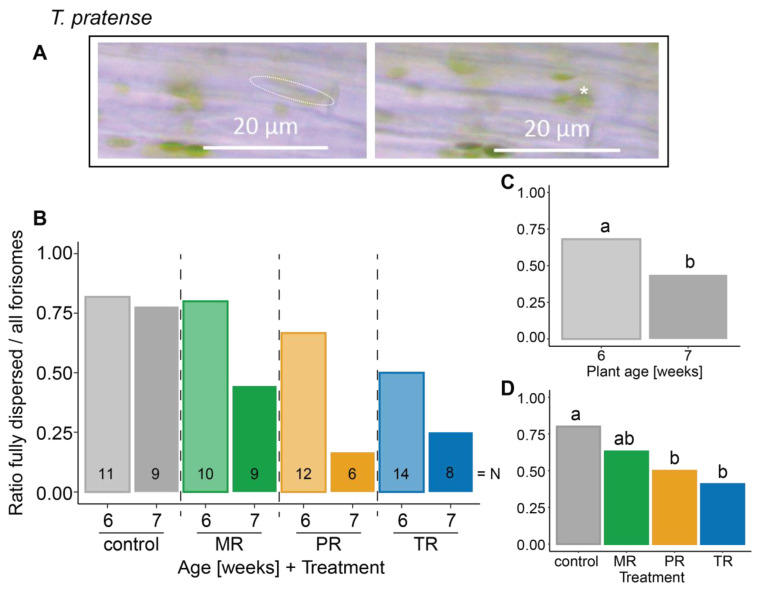
Influence of different *A. pisum* host races on the dispersion behavior of forisomes in the native host plant *T. pratense*. (**A**) *T. pratense* forisome in its condensed (left panel, dashed line) and its dispersed (right panel, putative location indicated by asterisk) state. Depicted are the ratios of fully dispersed forisomes to all observed forisomes after heat stimulation (**B**) in various host race treatments on plants at both ages, (**C**) in various plant ages (indicated by fill shade), and (**D**) in various aphid host race treatments. Different lower-case letters above bars indicate significant differences (*p* ≤ 0.05). Data were analyzed by a Bernoulli GLM. Further statistical information can be found in Table 1. Numbers in the bars indicate the total number of forisomes observed for the respective treatment (N). MR—*Medicago* host race (green); PR—*Pisum* host race (orange); TR—*Trifolium* host race (blue).

**Figure 3 plants-12-01888-f003:**
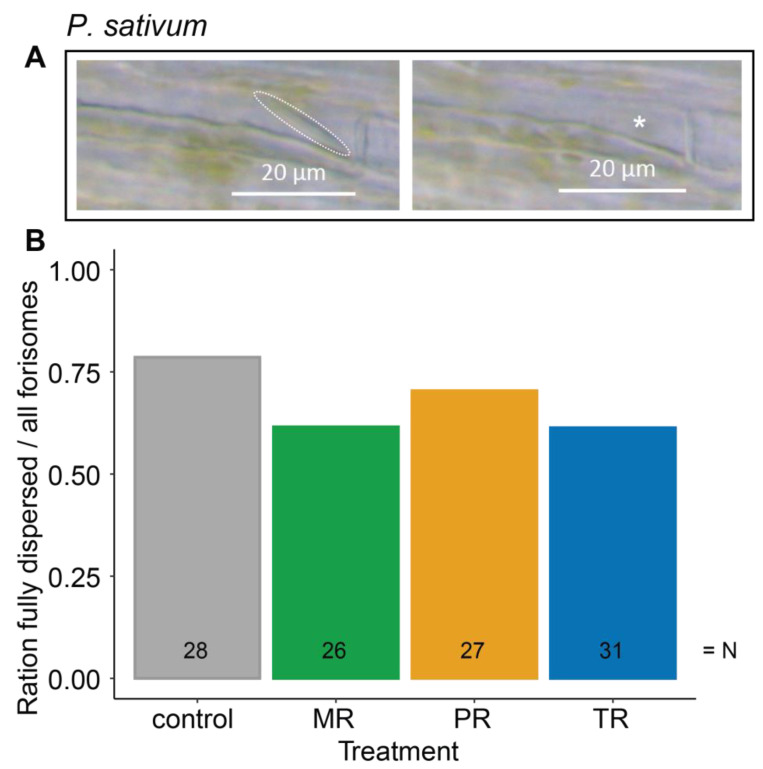
Influence of different *A. pisum* host races on the dispersion behavior of forisomes in the native host plant *P. sativum*. (**A**) *P. sativum* forisome in its condensed (left panel, dashed line) and its dispersed (right panel, putative location indicated by asterisk) state. (**B**) Depicted are the ratios of fully dispersed forisomes to all observed forisomes after heat stimulation in various aphid host race treatments. No significant differences were observed. Data were analyzed by a Bernoulli GLM. Further statistical information can be found in Table 1. Numbers in the bars indicate the total number of forisomes observed for the respective treatment (N). MR—*Medicago* host race (green); PR—*Pisum* host race (orange); TR—*Trifolium* host race (blue).

Forisome dispersion in *V. faba* was dependent on the pea aphid host race—plant age combination (*p* = 0.049; Figure 1B; Table 1). *Medicago* host race (MR) and *Pisum* host race (PR) aphids suppressed forisome dispersion on three-week-old *V. faba* compared to dispersion on uninfested control plants. On four-week-old plants, however, the dispersion rate was significantly higher in MR and PR infested plants compared to control plants. *Trifolium* host race aphids (TR) on *V. faba* did not alter forisome dispersion compared to the uninfested control, but dispersion was reduced on four-week-old plants compared to three-week-old plants.

Forisome dispersion in *T. pratense* [L.] was influenced by plant age (*p* = 0.031; Figure 2C) and the host race of the infesting aphid (*p* = 0.022; Figure 2D; Table 1) independently from each other (*p* = 0.516; Table 1). With increasing plant age, forisomes became less responsive to heat stimulation. Among host races, the non-native PR and particularly the native TR were able to suppress forisome dispersion in a significant manner compared to uninfested control plants.

On *P. sativum* [L.], pea aphids could not suppress forisome dispersion irrespective of whether native or non-native host races were used (*p* = 0.437; Figure 3B; Table 1).

The results for the three different plant species show that forisome responses depend significantly on the host race of the infesting aphid and plant age, as well as on the plant species.

### 2.2. Aphid Saliva Composition Depends on the Pea Aphid Host Race

The differential impact of the various aphid host races on forisome dispersion might be explained by differences in aphid saliva composition. In order to investigate this, we collected aphid watery saliva and analyzed it by gel electrophoresis. Visualization of native proteins revealed that PR was lacking a protein band at an apparent molecular weight of around 200 kDa (Figure 4A, arrowhead) that was present in the saliva of MR and TR. Further, we observed protein bands at an apparent molecular weight larger than 250 kDa for PR and TR, but not for MR (Figure 4A, asterisk).

When denatured samples from the same collections were analyzed by SDS-PAGE, the bands at molecular weights larger than 250 kDa in the PR and TR lanes could no longer be observed (Figure 4B), which indicates that these protein bands in the native protein gel likely arose from multimeric complexes. Additionally, the saliva of PR showed a distinct difference to that of MR and TR in that only a single band could be observed at 100 to 110 kDa instead of a double band (Figure 4B, arrowhead).

Carolan and colleagues previously investigated pea aphid saliva composition through SDS-PAGE and proteomics [39,49]. We correlated the proteins they reported with those in our gels based on their apparent molecular weight (compare Figure 4; Table 2). Previously reported proteins that appear to be present in our samples include a metalloprotease (130–150 kDa), a calcium-binding protein (regucalcin, 41 kDa), an angiotensin-converting enzyme (ACE, 100–115 kDa) and a putative sheath protein (130–150 kDa) [39,49]. Not all proteins could be found in all samples, such as one homologue of the ACE, which was only present in MR and TR saliva (Figure 4B, arrowhead, 100 kDa).

## 3. Discussion

In order to see whether pea aphid host races are able to suppress forisome dispersion in Fabaceae plants and therefore, SE clogging, we utilized heat stimulation by burning to trigger a strong Ca^2+^ influx into the SE [3,50]. An increase in Ca^2+^ concentration is the only reason for forisome dispersion [14]. If aphids are able to suppress forisome dispersion they consequently must be able to manipulate Ca^2+^ levels in SEs. Additionally, if pea aphid host races are able to suppress forisome dispersion in spite of such a strong trigger like heat stimulation, we hypothesize that they would also be able to suppress feeding-triggered Ca^2+^ influx and forisome dispersion.

Our results clearly show that suppression of heat-triggered forisome dispersion is not a general phenomenon in all interactions between legume host plants and their native pea aphid host races. Forisomes of different plant species reacted differently ranging from no alteration of forisome dispersion by aphid infestation as in *P. sativum* to obvious suppression of forisome dispersion by aphids as in *T. pratense*, especially but not solely by the native pea aphid host race. Whether forisome dispersion in *V. faba* was suppressed by aphids was dependent on the aphid host race and the age of the plant. These diverse dispersion patterns may be explained in different manners.

In order to fully disperse, forisomes have to come into contact with a certain amount of Ca^2+^ ions (Figure 1) [50]. This is the only stimulus which leads to forisome dispersion [14]. Thus, forisomes can be considered as Ca^2+^ sensors, and non- or only partially dispersed forisomes indicate insufficient amounts of Ca^2+^ in the SEs to trigger forisome dispersion [14,50].

### 3.1. Influence of the Damage Caused by Different Pea Aphid Host Races

Even though aphids cause little damage to the plant since they navigate their stylet through the apoplast, they regularly pierce cells to orient themselves and find the SEs [18]. This slight damage can still lead to local Ca^2+^ bursts [51], which may ultimately trigger forisome dispersion [4]. The higher the number of aphids penetrating a plant, or the higher the number of piercing events by one aphid, the greater the damage, and the more Ca^2+^ bursts should occur in plant cells. Following this logic, aphid-infested plants experience more damage and, consequently, more Ca^2+^ bursts than aphid-free control plants. In theory, this should lead to a higher proportion of dispersed forisomes. This was, however, not the case in our study (Figure 1, Figure 2 and Figure 3). On the contrary, especially in *T. pratense*, fewer forisomes dispersed after infestation with the native TR aphid (40.9%) compared to the control plants (80%, Figure 2). There was also the tendency that *T. pratense* infested with the native TR aphid showed less forisome dispersion (40.9%) than when infested with the non-native MR aphid (63.2%, Figure 2). If damage would have been the main driver in this interaction, native TR aphids would have caused less damage than MR aphids. However, the contrary is the case [31]. Previous experiments have shown that more aphids pierce SEs during a compatible plant—aphid interaction than during an incompatible interaction. Also, the time aphids of a compatible interaction spend during a compatible interaction navigating their stylets through the epidermis and mesophyll is longer than the time aphids spent during an incompatible interaction [31]. Thus, all in all, the observed forisome dispersion patterns cannot be explained by different degrees of damage caused by pea aphid host races, but must be attributed to host race-specific differences in physiology or behavior.

### 3.2. Possible Mechanisms of Forisome Dispersion Manipulation

The one and only way that forisomes change their conformation is via an increase of the Ca^2+^ concentration in the SEs [10,52]. Thus, since some aphid host races are able to suppress forisome dispersion, they must be able to suppress heat-induced Ca^2+^ level increases. There are several potential and not mutually exclusive ways of how Ca^2+^ levels in SEs and therefore forisome dispersion can be influenced by aphids: **1**. By manipulation of signal transduction and consequently altered Ca^2+^ influx into the SE. **2**. By manipulation of Ca^2+^ channels responsible for Ca^2+^ influx and efflux. **3**. By differences in Ca^2+^ levels among plant species. **4**. Through Ca^2+^-scavenging capacity of the aphid saliva. **5.** By the quality and quantity of aphid saliva injected into the plant.

#### 3.2.1. Signal Transduction and Forisome Dispersion

The Ca^2+^ influx into the cytosol of SEs that triggers forisome dispersion occurs through Ca^2+^ channels [53]. The Ca^2+^ channel opening can be mediated by electrophysiological reactions (elRs) [54,55]. Strong elRs along the plant vasculature can be triggered by heat stimulation (burning) of the leaf [3,14,50]. We have previously shown that forisome dispersion is triggered by variation potentials (VPs) and electropotential waves (EPWs), which are a combination of VP and action potential (AP) [55]. Since APs alone did not trigger forisome dispersion, VPs must be responsible for forisome dispersion. VPs, however, result from chemical components [56] or the propagation of pressure waves within the xylem vessels, which lead to local activation of mechano-sensitive Ca^2+^ channels of the SE plasma membrane [57,58,59]. Since the xylem is seldom pierced by nonmigrating parthenogenetic pea aphids [31], direct interference with the signal transduction along the xylem is quite unlikely. However, the decline in forisome dispersion with increasing plant age in *T. pratense* (from 68.1% to 43.7%, see Figure 2C) and aphid-free *V. faba* (from 73.7% to 55.6%, see Figure 1B) might result from changes in signal transduction. With increasing age, more lignin is deposited in the cell walls, increasing the rigidity of the vascular system [60]. This may negatively affect the hydraulic pressure wave necessary to transmit the VP and, hence, prevent VP-triggered forisome dispersion [61,62].

#### 3.2.2. Manipulation of Ca^2+^ Channels Responsible for the Influx and Efflux of Ca^2+^

Upon arrival of a given signal, ion channel opening is needed to transfer Ca^2+^ from the apoplast or internal storage, such as the endoplasmic reticulum, into the cytosol. Different Ca^2+^ channels are known so far in plants [53,63,64], for example GLUTAMATE RECEPTOR-LIKE proteins (GLR) which are activated through amino acid ligands and necessary for VP propagation [65]. One of these channels (GLR3.3) is implicated in plant defense [66] and GLR3.4 is phloem-located [67]. Both could, therefore, be a target of aphid manipulation [6]. The inhibition of GLRs may be achieved through low molecular weight amino acid analogs present in the aphid watery saliva.

After stimulus-triggered Ca^2+^ increases, the cytosolic Ca^2+^ concentration has to be returned to the resting levels in order for forisomes to recondense [68]. This offers an opportunity for native aphids to manipulate forisome dispersion since an increased efflux rate would also negatively impact Ca^2+^-dependent forisome dispersion. The saliva of MR was shown to contain regucalcin (41 kDa, see Figure 4 and Table 2), which is a Ca^2+^ binding protein that is known to activate Ca^2+^ pumps [69]. Hence, MR may accelerate the Ca^2+^ efflux from the SE and reduce forisome dispersion as seen in three-week-old *V. faba* (from 73.7% to 56.2%, see Figure 1B). Whether such alteration of Ca^2+^ import/export channels really takes place and whether it can influence forisome dispersion still needs to be investigated.

#### 3.2.3. Differences in Ca^2+^ Levels among Plant Species

Whereas the Ca^2+^ levels in SEs at resting conditions are at a universally low level of around 0.05 µM [50,70], Ca^2+^ levels differ between plant tissues and under different water potentials [71], and so might also differ between plant species in a way that affects the aphid’s ability to manipulate forisome dispersion. However, to date there is little information regarding the quantitative changes in Ca^2+^ levels in SEs [15,50]. It is known that forisomes of *V. faba* require 60–100 µM Ca^2+^ in order to disperse [50,72], and it is also known that Ca^2+^ concentrations required for forisome dispersion likely increase with forisome size [55]. Thus, since the forisomes of *P. sativum* are in general bigger than those of *T. pratense* [55], it is possible that the induced Ca^2+^ concentrations in SEs of *P. sativum* are higher than in those of *T. pratense*. Assuming that the saliva from some pea aphid host races has the Ca^2+^-scavenging capacity to suppress forisome dispersion in *T. pratense* (consistent with our results, forisome dispersion reduction PR 30%, TR 39.1%, Figure 2D), this capacity might not be enough to suppress the dispersion of the larger forisomes of *P. sativum* (consistent with no forisome dispersion reduction by aphids in our study, Figure 3B) with their potentially higher Ca^2+^ levels.

#### 3.2.4. Ca^2+^-Scavenging Capacity of the Aphid Saliva

It has been discussed for some time that aphids might be able to suppress forisome dispersion by the Ca^2+^-scavenging capacity of their saliva [37,38,68,73]. Ca^2+^-scavenging proteins in aphid saliva have been reported for *M. viciae* [38], as well as *A. pisum* [39,74]. However, when feeding on *P. sativum*, the scavenging capacity of the investigated pea aphid saliva does not seem to be large enough, since none of the pea aphid host races in our study were able to suppress heat-induced forisome dispersion (see Figure 3).

In *T. pratense*, on the other hand, heat-triggered forisome dispersion is differentially affected by pea aphid host races (80% control plants, 63.1% MR, 50% PR, 40.9% TR, see Figure 2D), which could at least be partly explained by variation in the Ca^2+^-scavenging capacity of the aphid saliva. It is likely that the Ca^2+^-scavenging capacity differs between the different pea aphid host races, since, as discussed above, the forisomes of the investigated host plants vary in their sizes. Due to forisomes in *P. sativum* being larger than those in *T. pratense* [55], it can be assumed that more Ca^2+^ ions are needed for full dispersion. Consequently, it is likely that the saliva of PR aphids has a higher Ca^2+^-scavenging capacity than the saliva of TR aphids. This might explain why the non-native PR could substantially supress forisome dispersion on *T. pratense* from 80% to 50% (see Figure 2D).

That the Ca^2+^-scavenging capacity of saliva differs between pea aphid host races is further supported by the differences in protein composition presented (Figure 4) and previous research showing variation in pea aphid saliva composition among host races [38,39,75]. So far, only a few Ca^2+^-scavenging proteins (regucalcin, ARMET) are known [39,49,69,74] and the function of the vast majority of saliva proteins still needs to be elucidated. More Ca^2+^-scavenging proteins may be found in the future whose occurrence differs among the pea aphid host races.

Since *V. faba* is the universal host plant for all pea aphid host races, we hypothesized that all host races would be able to suppress heat-induced forisome dispersion on this plant. Yet, we only observed suppression for the MR and PR aphids and only in younger plants (from 73.7% to 56.2% for MR and to 47.6% for PR, Figure 1B). The ratio of dispersed forisomes in MR- and PR-infested older plants was even higher (+31.9% for MR, +6.9% for PR) than in aphid free plants of the respective age. Previous reports indicated that the MR was also not able to reverse forisome dispersion in preflowering *V. faba* plants [42,48]. Such a pattern indicates that the Ca^2+^-scavenging capacity of aphid saliva by itself is likely not sufficient to explain forisome dispersion ability, implicating a stronger role for Ca^2+^ channel manipulation as discussed above.

#### 3.2.5. Quality and Quantity of Aphid Saliva

Most of the mechanisms for suppressing forisome dispersion discussed above can be reinforced if aphids inject more saliva into the plant. So far, we cannot quantify the amount of saliva injected by one aphid, but the amount of saliva injected into SEs depends also on the number of aphids feeding on a plant. Infestations by native host races result in more aphids reaching the phloem and feeding on the plant than infestations of non-native host races [31]. Since aphid feeding is associated with salivation into the plant [21,76], likely more saliva from native host races is injected into the host plant than from non-native host races. This assumption is supported by an aphid feeding study from Schwarzkopf and colleagues [31], which showed that nearly all TR aphids salivated into SEs on their native host plant *T. pratense* but only a low portion of non-adapted MR and PR aphids did so. This mechanism would nicely explain the strong suppression of forisome dispersion by the native TR on *T. pratense* from 80% to 40.9%) and the lower ability to supress forisome dispersion of the non-native MR (from 80% to 63.1%, see Figure 2D).

To further complicate matters, not only the amount of saliva secreted into a plant changes with plant species, but also the composition of the aphid saliva changes depending on the plant the aphid is feeding on [47,77]. This makes it even more difficult to pinpoint the precise mode of action involved in the suppression of forisome dispersion.

### 3.3. Closing Remarks

Since forisome-triggered SE occlusion depends on Ca^2+^ ions, Ca^2+^ scavenging and Ca^2+^ channel manipulation are two mechanisms that could act in concert to suppress forisome dispersion and thereby enable the different pea aphid host races to feed on their respective host plants. While Ca^2+^ scavenging prevents forisome dispersion as well as the opening of potential Ca^2+^-dependent channels in the SE, direct manipulation of the Ca^2+^ channels can prevent phloem defense responses even before the aphid pierces a given SE [4,42,64].

Regardless of the different mechanisms by which aphids might change forisome dispersion, we have to keep in mind that in this study we actually investigated the capacity of the aphids to suppress heat-induced forisome dispersion as a proxy for suppression of forisome dispersion induced by aphid feeding. Heat stimulation induces strong Ca^2+^ fluxes that may surpass feeding-triggered increases, and Ca^2+^ bursts and subsequent forisome dispersion might be much lower upon actual aphid infestation without heat treatment. Thus, if aphids are able to suppress forisome dispersion triggered by a heat stimulus, they might easily be able to suppress forisome dispersion triggered during feeding. Aphids that were not able to suppress heat-induced forisome dispersion in this study may be able to more readily cope with the lower Ca^2+^ concentrations in SEs that might occur during normal feeding. The fact that some host races are able to suppress heat-triggered forisome dispersion illustrates that aphids can counter this phloem defense mechanism and paves the way for future studies on the underlying mechanisms.

## 4. Materials and Methods

### 4.1. Plant and Aphid Cultivation

The experiments were conducted on *Pisum sativum* cultivar (cv) ‘Baccara’, *Trifolium pratense* cv ‘Dajana’ and *Vicia fab*a cv ‘The Sutton’ since they represent either native host plants (*P. sativum* and *T. pratense*) or the universal host plant (*V. faba*) for pea aphid host races. This allowed us to investigate compatible and incompatible combinations of aphids and plants.

The plants were cultivated in 10 cm diameter plastic pots with a standardized soil mixture of Klasmann Tonsubstrat and Klasmann Kultursubstrat TS1 (proportion 7:20; Klasmann-Deilmann GmbH, Geeste, Germany). The temperature of the growth chamber was maintained between 20 and 22 °C and the relative humidity between 60 and 70%. Long day conditions (L16:D8) were used with an irradiance level of 100 to 150 µmol m^−2^ s^−1^ (Fluora lamps, Osram GmbH, Munich, Germany).

All *A. pisum* clones used were free of facultative endosymbionts and were previously obtained from ampicillin-treated naturally occurring clones. More information regarding these pea aphid clones can be found in [78,79]. The *A. pisum Medicago* host race (MR; clone ID218—obtained from L84), *Pisum* host race (PR; clone ID212—obtained from P123) and *Trifolium* host race (TR; clone ID210—obtained from YR2) were reared separately in small tents (Bugdorm; MegaView Science Co., Ltd., Taiwan) on *V. faba* under the plant growth conditions listed above.

### 4.2. Experimental Set-Up

Studies of local influences on forisome dispersion were executed on *P. sativum* (4 weeks), *T. pratense* (6–7 weeks) and *V. faba* (3–4 weeks) in their vegetative state before flowering. Aphid-free plants served as controls. Each plant species was infested separately with all three pea aphid host races. Whole plants were infested with around 150 to 200 apterous, parthenogenetic aphids of mixed age, which were allowed to feed for 48 h. To prevent the escape of aphids, all experimental plants were caged in tents according to aphid host race.

Plants were handled in a manner so as not to dislodge feeding aphids from the plants. For in vivo observations of forisome reactions, the phloem was exposed by removing the cortical cell layers from the lower side of the midvein with a razor blade as described previously [13,80]. The leaf was then fastened upside down onto an objective slide with double-sided adhesive tape. The exposed tissue was immersed in physiological bathing medium (2 mM KCl, 1 mM CaCl_2_, 1 mM MgCl_2_, 50 mM mannitol, 2.5 mM MES/NaOH buffer, pH 5.7) and allowed to incubate for one hour. The integrity of the tissue was subsequently analyzed with a light microscope (AXIO Imager.M2, Zeiss, Jena, Germany) using a 40 x water immersion objective (W-N Achroplan, Zeiss).

To trigger forisome dispersion, a heat stimulus was applied for 2 s with a match to the tip of the observation leaflet. For *V. faba*, this was leaf number 4, for *P. sativum* the second youngest, mature leaflet, and for *T. pratense* the central leaflet (compare Figure 5). In accordance with previous research [55], we limited the range of the distance from the stimulus to the observation site to 2 to 3 cm in *V. faba*, 1.5 to 3 cm in *T. pratense* and 1 to 2.5 cm in *P. sativum*. Each plant was used once and only plants with intact SEs and forisomes located on the downstream (basal) side of the SE were considered for experiments. It was recorded whether the forisomes dispersed, did not disperse or partially dispersed, which was counted as unsuccessful dispersion. Heat triggered forisome reactions were traced with a color camera (AXIOCAM 503 color, Zeiss) and micrographs processed with the ZEN (blue edition) software (Zeiss). The figures presented were compiled with Adobe Illustrator CS5 (Dublin, Ireland).

### 4.3. Aphid Saliva Collection and Preparation

On average, 100 aphids of all ages from each host race were used to collect saliva. For this, 1 mL sterile artificial diet (100 mM aspartic acid, 100 mM methionine, 100 mM serine, 15% sucrose, pH 7.2 (KOH) [38]) was plated between two Parafilm (Beemis Company Inc., Neenah, IW, USA) layers, which were covered by a cage to prevent the aphids from escaping. After 24 h, the aphids were removed and the remaining diet collected.

In order to concentrate saliva proteins larger than 3 kDa and cleanse the concentrate of surplus sugar and amino acids, the protocol of Will and colleagues [38] was complemented with multiple washing steps. The saliva-diet suspension of at least 6 collections was transferred to a Vivaspin 20 concentrator (3 kDa cutoff, PES; Sartorius, Göttingen, Germany) and the volume increased to 20 mL with 10 mM Tris-HCl (pH 7.3). After 1:20 h centrifugation at 4 °C and 4000× *g*, another 5 mL Tris-HCl were added and the centrifugation step was repeated. The remaining concentrate was again filled up with Tris-HCl to a final volume of 20 mL and spun for another 2 h. The supernatant was transferred to a Vivaspin 500 column (3 kDa cutoff, PES; Sartorius). After one hour of centrifugation at 4 °C and 4000× *g*, 500 µL Tris-HCl (pH 7.3) were added and the samples were centrifuged overnight. Afterwards 200 µL 10 mM Tris-HCl (pH 7.3) were added to the supernatant and the samples were centrifuged another five to six hours at 4000× *g* and one to two hours at 10,000× *g*. The retained saliva samples (ca. 15 to 20 µL each) were stored at −80 °C until further usage.

### 4.4. Determination of Protein Concentration and Saliva Compositions

The protein concentration of each concentrated saliva sample was determined by adding 2 µL to 50 µL Bradford solution (QuickStart^TM^ Bradford Prot Assay; Bio-Rad Laboratories, Hercules, CA, USA) in a 96 well plate (PS, F-bottom, clear; Greiner bio-one, Frickenhausen, Germany). After 10 min incubation, the absorption was measured at 595 nm with a Tecan multi-well reader (infinite M200; Tecan Austria GmbH, Grödig/Salzburg, Austria) using optimal gain and a number of 25 flashes.

To examine the saliva composition, 300 ng protein per host race were applied to a polyacrylamide gel. As a marker 1 µL PageRuler^TM^ Plus Prestained Protein Ladder (Thermo Fisher Scientific, Pittsburgh, PA, USA) was utilized. For native gel electrophoresis, the samples were separated on an 8% Tris-glycine gel (Invitrogen^TM^ Novex^TM^ WedgeWell^TM^; Thermo Fisher Scientific) submerged in a 25 mM Tris, 192 mM glycine running buffer. Native gel electrophoresis was executed with a voltage of 150 V for 1:10 h. For non-native gel electrophoresis, the samples were denatured for 5 min at 90 °C in the presence of SDS loading dye (6x; G-Biosciences, St. Louis, MO, USA) with 2.5% β-mercaptoethanol. SDS-PAGE was executed by separating the proteins on a pre-cast Mini-PROTEAN TGX gel (AnykD, 12 well, 20µL/well; Bio-Rad Laboratories, Feldkirchen, Germany) for 1:10 h at 100 V submerged in running buffer (25 mM Tris, 192 mM glycine, 0.1% SDS, pH 8.3).

After gel electrophoresis the gel was fixed by steeping it four times in fixing solution (40% EtOH, 10% acetic acid) for 15 min. Three 20 min washing steps (30% EtOH) were followed by three 10 min rinsing steps (ddH_2_O). Sensitizer (0.02% Na_2_S_2_O_3_) was applied for one minute and the gel subsequently rinsed three times with ddH_2_O. The gel was incubated in staining solution (0.2% AgNO_3_, 0.02% formaldehyde) for 20 min and then rinsed three times. The developer (3% Na_2_CO_3_, 0.04% formaldehyde) was added to the gel and immediately removed after the staining was as strong as required. After washing two times with ddH_2_O, a stop solution (5% acetic acid) was added for 5 min. Gels were stored in 1% acetic acid. Pictures were taken with a Lumix Digital Camera DMC-FZ200 (Panasonic, Wiesbaden, Germany) and figures compiled with Adobe Illustrator CS5.

### 4.5. Statistics

To investigate the influence of different *A. pisum* host races on forisome dispersion, the dispersion behavior was categorized as either fully dispersed or not fully dispersed. In case multiple forisomes were observed in one plant, one forisome was selected at random with the help of R. The influence of the *A. pisum* host race, plant age (both used as fixed explanatory factors) and distance between heat stimulus and observation site (used as fixed continuous explanatory variable) on the ratio of fully dispersed forisomes was analyzed per plant with a Bernoulli generalized linear model (GLM). To determine the *p*-values of the main explanatory factors, the explanatory factors were removed one after another from the model and the simpler models were compared to the more complex model with an analysis of deviance test [81]. In case of significant differences, factor level reduction was applied to detect differences between treatments [82]. All statistical analyses were done in R version 4.2.1 [83].

## Data Availability

Data can be obtained upon request from the corresponding authors.

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
