# Peer review of "Pea Aphid (Acyrthosiphon pisum) Host Races Reduce Heat-Induced Forisome Dispersion in Vicia faba and Trifolium pratense"

_plants, 2023, doi:10.3390/plants12091888_

Round 1

Reviewer 1 Report

The advance of knowledge in plant-aphid interaction biology focusing on sieve element based forisome dispersion in host resistance have merit to publish for readers. 

Author Response

We are pleased that the reviewer liked our manuscript. After the revision of our manuscript we have carefully spell checked it.

Reviewer 2 Report

Paulmann et al. described an interesting phenomenon that pea aphid damage can suppress the forisome dispersion induced by heating. They also found that the salivary proteins from different pea aphids host races have different protein components. Overall, the study could be useful for understanding the mechanism of how aphid interact with host plants.

I have several concerns and hope the authors can address them:

(1) The title can be modified to make sure it fits the contents better. for instance, Pea aphid (Acyrthosiphon pisum) feeding affects the heat-induced forisome dispersion. Since the induction of forisome was also observed in the current study.

(2) It would be better to provide more detailed background information on the three pea aphid races used in the introduction part. I am confused why do you call them 'races', is it because they are collected from different plant species? Or because they showed specialized fitness/feeding preference to different host plants? And why do you select these races? How about their performances on the non-preferred hosts? Is it possible that the different regulation of forisome dispersion is caused by the fact that different aphids show significant difference in feeding preference and thus cause different level of damages to host/non-preferred host plants? If so, then checking the damage level wound make sense, and the feeding style, but not the chemical induction (Ca2+), might be the reason for the difference of forisome dispersion.   

(3) in the discussion part, the authors talked a lot about the Ca2+ ions. While they did not carry out any measurement of this ion, it is not safe to link this ion directly to the findings in their results. So please remove the content from line 409 to line 649, and line 674 to line 692. And rewrite the Summary part by focusing on what the data in the current study tells.

(4) If you have captured the pictures for plant tissues when analyzing the forisome dispersion, please also select those representative ones and show them beside Fig 1, 2 and 3. So the readers can see what happened more directly.

(5) I am worried if the aphid saliva composition is logically linked to the forisome dispersion suppression. Otherwise the authors need to provide direct evidences that the salivary protein mixture is responsible for the differentiated (and potential) regulation of plant responses. Probably it is better make this data as supplementary materials.

Author Response

I have several concerns and hope the authors can address them:

(1) The title can be modified to make sure it fits the contents better. for instance, Pea aphid (Acyrthosiphon pisum) feeding affects the heat-induced forisome dispersion. Since the induction of forisome was also observed in the current study.

Answer: We modified the title to “Pea aphid (Acyrthosiphon pisum) host races reduce heat-induced forisome dispersion in Vicia faba and Trifolium pratense”.

(2) It would be better to provide more detailed background information on the three pea aphid races used in the introduction part. I am confused why do you call them 'races', is it because they are collected from different plant species? Or because they showed specialized fitness/feeding preference to different host plants? And why do you select these races? How about their performances on the non-preferred hosts? Is it possible that the different regulation of forisome dispersion is caused by the fact that different aphids show significant difference in feeding preference and thus cause different level of damages to host/non-preferred host plants? If so, then checking the damage level wound make sense, and the feeding style, but not the chemical induction (Ca2+), might be the reason for the difference of forisome dispersion.

Answer: We have now added more detail to the paragraph in the introduction about the different pea aphid host races (line 67 ff). We now explain that the pea aphid host races are genetically different. We also mention that they survive and reproduce only or at least significantly better on their respective host plant and on the universal host plant compared to other non-host legume plants.

The aphid races were chosen because their respective host plants are relatively easy to cultivate (compared to pea aphid host plants like Cytisus scoparius or Ononis spinosa). Additionally, quite a few experiments have already been conducted with these aphid-plant combinations so that we can interpret the findings of the current work much better. These experiments are mentioned also in the introduction in the paragraph on pea aphids.

More information on the selected clones can be found in Peccoud, J., et al. (2009). "Post-Pleistocene radiation of the pea aphid complex revealed by rapidly evolving endosymbionts." Proceedings of the National Academy of Sciences of the United States of America 106(38): 16315-16320 and Simon, J.C.; et al. (2011). “Facultative symbiont infections affect aphid reproduction.” PLoS One 6, e21831. Both papers are now mentioned in the Material and Methods section.

The reviewer is right, we did not discuss the different feeding preferences and their potential role for forisome dispersion. We have now inserted a paragraph which discusses in detail the possible role of different damage levels caused by the different pea aphid host races. However, the ultimate reason why a forisome disperses or not is always the Ca2+ concentration. Without Ca2+ ions, forisomes do not disperse. Thus, even the damage caused by aphid infestation is linked to the Ca2+ levels in the sieve elements.

(3) in the discussion part, the authors talked a lot about the Ca2+ ions. While they did not carry out any measurement of this ion, it is not safe to link this ion directly to the findings in their results. So please remove the content from line 409 to line 649, and line 674 to line 692. And rewrite the Summary part by focusing on what the data in the current study tells.

Answer: We don’t know which lines the reviewer wants to be removed since the line numbers mentioned refer to the Material and Methods section in our version, which is most likely not what the reviewer intended.

The reviewer is right, our whole discussion is about Ca2+ ions even though we have not actually measured Ca2+ concentrations. We would love to measure Ca2+ concentrations in sieve elements but in non-model plants which cannot be genetically transformed this is not really possible to date. However, forisomes only disperse if Ca2+ concentrations are high enough. Therefore, forisomes can be considered as natural Ca2+ bioindicators (van Bel, A. J. E., et al. (2014)). Thus, if in a certain legume plant forisomes show different dispersion behavior in different treatments, Ca2+ concentrations must be different between these treatments.

Since the one and only way that forisomes change their conformation is via the increase of the Ca2+ concentration in the sieve-elements, we have discussed the various mechanisms that can influence Ca2+ concentrations.

(4) If you have captured the pictures for plant tissues when analyzing the forisome dispersion, please also select those representative ones and show them beside Fig 1, 2 and 3. So the readers can see what happened more directly.

Answer: We have now added pictures of forisomes to Figures 1 – 3, so that the readers can see what we have investigated.

(5) I am worried if the aphid saliva composition is logically linked to the forisome dispersion suppression. Otherwise the authors need to provide direct evidences that the salivary protein mixture is responsible for the differentiated (and potential) regulation of plant responses. Probably it is better make this data as supplementary materials.

Answer: Current knowledge gives strong links between aphid saliva composition and the regulation of plant responses. As an example, RNAseq analyses revealed that the majority of genes which are promising candidates for the genetic basis of host–plant specialization in the pea aphid complex encode salivary proteins (Nouhaud, P. et al.: Identifying genomic hotspots of differentiation and candidate genes involved in the adaptive divergence of pea aphid host races. Mol Ecol 2018). There is already direct evidence that aphid saliva can suppress forisome dispersion in vitro (Will, T.et al.: Molecular sabotage of plant defense by aphid saliva. PNAS 2007). And regucalcin is one saliva protein which has a Ca2+ scavanging capacity (Carolan et al.: The secreted salivary proteome of the pea aphid Acyrthosiphon pisum characterised by mass spectrometry. Proteomics 2009). Thus, we think that there is a strong link between aphid saliva composition and supression of forisome dispersion, and so feel that it would be useful for the reader to have these data in the main manuscript.

Reviewer 3 Report

Dear colleagues,

This article is concerning a research work entitled “Pea aphid (Acyrthosiphon pisum) host races suppress forisome dispersion in Vicia faba and Trifolium pratense, by

Maria K. Paulmann, Linus Wegner, Jonathan Gershenzon, Alexandra C. U. Furch, Grit Kunert. As new data, I recommend it for an international audience in this journal, however several points have to be considered by the authors, and a major revision is requested, mainly for a re-organisation of this interesting paper.

Please notice that in order to bring a broad audience to this paper and to this journal, for specialists and non-specialists, the three major points of my comments (at the beginning) are very important (mandatory…) for a suitable value of the article. Minor points are also enhanced at the end of this review.

I deeply hope to see this good article published soon,

The three major points are:

1-1      The two following points are complementary. The first point embarrassing me is that the interesting discussion relates too shortly to the present results (almost no present results sustain this part) and the paper looks not equilibrated with this title. The solution would be to: 1/restrict the results part to the only description of values and etc…, and remove all sentences which belong actually to the discussion part (especially for 2.2, see also the other points of your results); in these results, add some values of the tables to be more easily readable, especially for non-specialists; 2/ make two sub-parts  in the discussion; one called "discussion" using precise values of your results and not only summarized remarks; 3/ then put all other (very interesting, however very long compared with the present results) remarks in a sub-part called "considerations on ...” or some sub-title like this (and change the title of your paper accordingly as in the present text it does not correspond to the paper).

2-2     The second point is that in details: 1/ the first paragraph of the discussion is actually an introduction (?); 2/ at the end of the introduction, from the sentence "we found...", it seems to belong to the summary or conclusion? 3/ at the end of point 3., the so-called summary is actually a "conclusion and perspectives"?

3-3     References already taken in account by the authors are of real interest, however checking briefly in the word of science WOS and scilit (from mdpi) with the key-words of the abstract, other references appear, and they should be once more selected and used (if relevant…) in order to provide a larger view of this interesting research. Among these are the followings:

[1-7]

1.         Knoblauch, M.; Froelich, D.R.; Pickard, W.F.; Peters, W.S. SEORious business: structural proteins in sieve tubes and their involvement in sieve element occlusion. Journal of Experimental Botany 2014, 65, 1879-1893, doi:10.1093/jxb/eru071.

2.         Liu, Y.; Peters, W.S.; Froelich, D.R.; Howell, A.H.; Mooney, S.; Evans, J.E.; Hellmann, H.A.; Knoblauch, M. Aspartate Residues in a Forisome-Forming SEO Protein Are Critical for Protein Body Assembly and Ca2+ Responsiveness. Plant & Cell Physiology 2020, 61.

3.         Noll, G.A.; Muller, B.; Ernst, A.M.; Ruping, B.; Twyman, R.M.; Prufer, D. Native and artificial forisomes: functions and applications. Applied Microbiology and Biotechnology 2011, 89, 1675-1682, doi:10.1007/s00253-011-3117-6.

4.         Rose, J.; Brand, I.; Bilstein‐Schloemer, M.; Jachimska, B.; Twyman, R.M.; Prüfer, D.; Noll, G.A. The Ca2 response of a smart forisome protein is dependent on polymerization. Protein Science 2021, 31.

5.         Rose, J.; Müller, B.; Groscurth, S.; Giese, J.; Eirich, J.; Finkemeier, I.; Twyman, R.M.; Prüfer, D.; Noll, G.A. The functionality of plant mechanoproteins (forisomes) is dependent on the dual role of conserved cysteine residues. International Journal of Biological Macromolecules 2021, 193.

6.         Smallwood, C.; Podlevsky, J.; Ryan, E.; Snow, T. Engineering Forisome Scaffolds: Elucidating Spatial Self-Assembly Patterning of Bio-inorganic Complexes; 2021.

7.         Srivastava, V.K.; Tuteja, R.; Tuteja, N. Calcium-energized motor protein forisome controls damage in phloem: potential applications as biomimetic "smart'' material. Critical Reviews in Biotechnology 2015, 35, 173-183, doi:10.3109/07388551.2013.823598.

 Minor points are:

1 As I am involved in taxonomy I am very sensible to correct taxa names, so put all latin names throughout the text with names of the author(s) of all taxa cited at least the first time they appear in the text (from the introduction): for plants use international Plant Names Index (IPNI) https://www.ipni.org/), or equivalent; for cultivars if there is no author’s name(s), put the reference where this name appears firstly in the literature. For animals use zoobank.org, organismnames.com, or equivalent;

2 in the introduction first paragraph, what are actually the "two mechanisms"?

3 for figure 2, indicate in the caption the meaning of a and b in B, of a ab b in C.; use colors for the diagrams of figs. 2-3, in order to be more attractive for the readers;

4 for figure 4, indicate in the caption the meaning of the numbers of the left parts of A and B;

5 for scheme 1, the photos in the middle are too small and hardly visible;

6 in 4.1 or at the end of the introduction, indicate briefly the reason(s) of the selection of these plants.

Author Response

The three major points are:

1-1 The two following points are complementary. The first point embarrassing me is that the interesting discussion relates too shortly to the present results (almost no present results sustain this part) and the paper looks not equilibrated with this title. The solution would be to: 1/restrict the results part to the only description of values and etc…, and remove all sentences which belong actually to the discussion part (especially for 2.2, see also the other points of your results); in these results, add some values of the tables to be more easily readable, especially for non-specialists;

Answer: We agree with the reviewer that we have put some explanatory information into the Results section. We have now removed some of this to make it easier for non-specialists to understand the results. We have also now added the sizes of the proteins mentioned in the table now in the text.

2/ make two sub-parts  in the discussion; one called "discussion" using precise values of your results and not only summarized remarks; 3/ then put all other (very interesting, however very long compared with the present results) remarks in a sub-part called "considerations on ...” or some sub-title like this (and change the title of your paper accordingly as in the present text it does not correspond to the paper).

Answer: We now restructured the discussion. In the first part we discuss our findings in the light of different degrees of damage caused by the three pea aphid host races. The second part is more speculative and discusses the possible ways how aphids might manipulate Ca2+ levels.

2-2 The second point is that in details: 1/ the first paragraph of the discussion is actually an introduction (?); 2/ at the end of the introduction, from the sentence "we found...", it seems to belong to the summary or conclusion? 3/ at the end of point 3., the so-called summary is actually a "conclusion and perspectives"?

Answer 1: The reviewer is right and we have removed most of the first paragraph of the discussion and left only a few introductory words to make the entry in the discussion a bit easier.

Answer 2: The reviewer is right that a short summary of the findings is present at the end of the introduction. However, this is requested by the journal so we have left it.

Answer 3: According to the suggestion of the reviewer we have now renamed the Summary as “Conclusion and perspectives”

3-3 References already taken in account by the authors are of real interest, however checking briefly in the word of science WOS and scilit (from mdpi) with the key-words of the abstract, other references appear, and they should be once more selected and used (if relevant…) in order to provide a larger view of this interesting research. Among these are the followings:

[1-7]

  1. Knoblauch, M.; Froelich, D.R.; Pickard, W.F.; Peters, W.S. SEORious business: structural proteins in sieve tubes and their involvement in sieve element occlusion. Journal of Experimental Botany 2014, 65, 1879-1893, doi:10.1093/jxb/eru071.
  2. Liu, Y.; Peters, W.S.; Froelich, D.R.; Howell, A.H.; Mooney, S.; Evans, J.E.; Hellmann, H.A.; Knoblauch, M. Aspartate Residues in a Forisome-Forming SEO Protein Are Critical for Protein Body Assembly and Ca2+ Responsiveness. Plant & Cell Physiology 2020, 61.
  3. Noll, G.A.; Muller, B.; Ernst, A.M.; Ruping, B.; Twyman, R.M.; Prufer, D. Native and artificial forisomes: functions and applications. Applied Microbiology and Biotechnology 2011, 89, 1675-1682, doi:10.1007/s00253-011-3117-6.
  4. Rose, J.; Brand, I.; Bilstein‐Schloemer, M.; Jachimska, B.; Twyman, R.M.; Prüfer, D.; Noll, G.A. The Ca2 response of a smart forisome protein is dependent on polymerization. Protein Science 2021, 31.
  5. Rose, J.; Müller, B.; Groscurth, S.; Giese, J.; Eirich, J.; Finkemeier, I.; Twyman, R.M.; Prüfer, D.; Noll, G.A. The functionality of plant mechanoproteins (forisomes) is dependent on the dual role of conserved cysteine residues. International Journal of Biological Macromolecules 2021, 193.
  6. Smallwood, C.; Podlevsky, J.; Ryan, E.; Snow, T. Engineering Forisome Scaffolds: Elucidating Spatial Self-Assembly Patterning of Bio-inorganic Complexes; 2021.
  7. Srivastava, V.K.; Tuteja, R.; Tuteja, N. Calcium-energized motor protein forisome controls damage in phloem: potential applications as biomimetic "smart'' material. Critical Reviews in Biotechnology 2015, 35, 173-183, doi:10.3109/07388551.2013.823598.

Answer: We appreciate the work of the referee and are aware of these interesting publications. But we decided not to cite the mentioned literature. The first publication is a review about sieve element occlusion, and forisomes are just one part of it. Therefore, we decided to include the original literature (17 publications) about forisomes mentioned in the review but not the review itself. Additionally, since the publication of the review in 2014 more knowledge was gained and open questions mentioned and discussed in the review are now answered, we also cited more recent publications in our manuscript.

Instead of publication 3, 4 and 5 we cited the most current review of this group (Noll et al. 2022). The other publications deal either with forisomes as smart materials or with the question how forisomes sense Ca2+ ions. However, both topics go far beyond the scope of the manuscript. Therefore, we did not cite them.

Minor points are:

  1. As I am involved in taxonomy I am very sensible to correct taxa names, so put all latin names throughout the text with names of the author(s) of all taxa cited at least the first time they appear in the text (from the introduction): for plants use international Plant Names Index(IPNI) https://www.ipni.org/), or equivalent; for cultivars if there is no author’s name(s), put the reference where this name appears firstly in the literature. For animals use zoobank.org, organismnames.com, or equivalent;

Answer: We have now added the author names to the species names when they appear for the first time in the main text.

  1. in the introduction first paragraph, what are actually the "two mechanisms"?

Answer: We added numbers to the text to make clear what the two mechanisms are. The text is now: “Plants have developed two main mechanisms to prevent the loss of phloem sap in response to mechanical damage [3] or phloem feeding insects [4-6]. 1) Long-term occlusion of SEs is achieved by the calcium-ion (Ca2+)-triggered accumulation of callose at the sieve pores and the plasmodesmata, while 2) short-term occlusion is achieved by phloem proteins (P-proteins) [3,7-11].”

  1. for figure 2, indicate in the caption the meaning of a and b in B, of a ab b in C.; use colors for the diagrams of figs. 2-3, in order to be more attractive for the readers;

Answer: We assume that the reviewer means the lower-case letters present in figure 1 and 2. We have now rephrased the figure caption to clarify the meaning of the letters. We also colored figures 1 to 3 as the reviewer requested.

  1. for figure 4, indicate in the caption the meaning of the numbers of the left parts of A and B;

Answer: We assume the reviewer must have overlooked the sentence in the figure caption that indicates the meaning of the numbers. We have now rephrased the caption to make it clearer: “Lane 1 (M): protein ladder (PageRulerTM Plus Prestained Protein Ladder). The corresponding protein sizes are given in kDa on the left side of each figure panel in lane ‘M’.”

  1. for scheme 1, the photos in the middle are too small and hardly visible;

Answer: We have now resized the scheme and made the photos bigger. We hope that the photos are now better visible.

  1. in 4.1 or at the end of the introduction, indicate briefly the reason(s) of the selection of these plants.

Answer: We have chosen these plant species because Pisum sativum is the native host plant to the Pisum race of the pea aphid, Trifolium pratense is the native host of the Trifolium race of the pea aphid, and Vicia faba is the universal host plant for all host races of the pea aphid. Thus, with the combination of the selected pea aphid host races, we have plant-aphid combinations that are compatible and those that are incompatible. Additionally, a few investigations have already been carried out on these plant species as well as on these plant species – aphid combinations. This gave us the possibility to interpret our findings in the light of known aphid feeding behavior or plant responses towards aphids.

A sentence has now been added to the Material and Methods section to give a rationale for the plant selection.

Round 2

Reviewer 2 Report

The authors have replied most of my suggestion. However, I insist that the Ca2+ related discussions is over stated since the current study did not perform any experiment on Ca2+. So the section “ 3.2. Possible mechanisms of Ca2+ level manipulation” should be reduced to maximum one paragraph.

Author Response

  • The reviewer is right, that we did not perform any experiment on Ca2+. This has two main reasons: First of all, precise measurements of Ca2+ concentration in specific cell types, such as sieve elements, are not possible unless the plants are genetically transformed, but sadly these legumes are not amenable to transformation to date. Second, we can use the forisomes themselves to measure Ca2+ levels since forisomes only change their conformation if a certain Ca2+ level threshold is reached. They are therefore useful Ca2+ sensors. In cases where forisomes are dispersed, the Ca2+ level must have been higher than in cases were forisomes are not dispersed. Thus, we can say that we did make an indirect determination of Ca2+ levels by observing the forisomes.
  • The reviewer feels that our detailed discussion about Ca2+ levels is too long. Since forisomes only disperse if a certain Ca2+ level is reached, and since we found that some aphid host races were able to suppress heat-induced forisome dispersion, it is clear that aphids somehow affect Ca2+ levels in sieve elements. To explain our findings adequately, we thus have to discuss how aphids might influence Ca2+ levels in sieve elements. So far, several physiological mechanisms are known about how Ca2+ levels are regulated (signal transduction initiates Ca2+ influx, membrane channels are responsible for Ca2+ influx into cells and efflux out of cells). However, how aphids or other herbivores might interfere with these mechanisms is largely unknown. Thus, we think it is very useful to our readers if we speculate about the different mechanisms aphids might use to manipulate Ca2+ levels. In order to make this more comprehensible especially for readers not deeply familiar with the topic, we have split this part of the discussion, giving a short paragraph for each possibility. It would seem to be much less clear if we combined this information into a single paragraph.
  • In trying to follow the reviewer’s comment, we have carefully checked the discussion and have now included a rationale for why we discuss Ca2+ levels in detail. We have also condensed this part of the discussion by keeping only the most important points.

Reviewer 3 Report

 Dear authors,

Good afternoon,

I read the corrected version of this article with interest, however I still maintain my first major point, not clearly corrected and/or argumented.

 Yours sincerely,

Actually I cannot see clearly the two parts evocated in the author’s notes by the authors (“We now restructured the discussion”). The discussion of this corrected version has a kind of introduction (?), then .3.1, then a very long 3.2 devoted to the ion Ca not present in the results. If the two first parts are now related with the present results as claimed in the author’s note, no values or etc from these results appear in your corrected version (there are “just” sentences with general summarized ideas; for instance for my teachings with my students I know that efficient discussion has to be sustained by clearly explained results with values and etc).

 In this respect, the discussion part is still too unconnected with the results parts, this version is still not equilibrated (Ca ion) and the title still not corresponding to the paper.

Author Response

  • The reviewer is right, we still had some introductory sentences at the beginning of the discussion. We have removed most of these and have kept just one introductory sentence to assure an easy entrance into the discussion.
  • The reviewer is right, that we did not perform any experiment on Ca2+. This has two main reasons: First of all, precise measurements of Ca2+ concentration in specific cell types, such as sieve elements, are not possible unless the plants are genetically transformed, but sadly these legumes are not amenable to transformation to date. Second, we can use the forisomes themselves to measure Ca2+ levels since forisomes only change their conformation if a certain Ca2+ level threshold is reached. They are therefore useful Ca2+ sensors. In cases where forisomes are dispersed, the Ca2+ level must have been higher than in cases were forisomes are not dispersed. Thus, we can say that we did make an indirect determination of Ca2+ levels by observing the forisomes.
  • The reviewer is right, we presented data in the Results section and did not mention it again in the Discussion section. We have now added important data to the Discussion section to make it easier to grasp.
  • The reviewer feels that our detailed discussion about Ca2+ levels is too long. Since forisomes only disperse if a certain Ca2+ level is reached, and since we found that some aphid host races were able to suppress heat-induced forisome dispersion, it is clear that aphids somehow affect Ca2+ levels in sieve elements. To explain our findings adequately, we thus have to discuss how aphids might influence Ca2+ levels in sieve elements. So far, several physiological mechanisms are known about how Ca2+ levels are regulated (signal transduction initiates Ca2+ influx, membrane channels are responsible for Ca2+ influx into cells and efflux out of cells). However, how aphids or other herbivores might interfere with these mechanisms is largely unknown. Thus, we think it is very useful to our readers if we speculate about the different mechanisms aphids might use to manipulate Ca2+ levels. In order to make this more comprehensible especially for readers not deeply familiar with the topic, we have split this part of the discussion, giving a short paragraph for each possibility. It would seem to be much less clear if we combined this information into a single paragraph.

Round 3

Reviewer 2 Report

I still think the discussion part is not scientifically sound

Author Response

The response can be find in the answer to the academic editor.

Reviewer 3 Report

Dear authors, 

 I read again this interesting manuscript, still not equilibrated between the different parts.

1-the abstract does not correspond entirely to the conclusion.

2-the title does not include the important discussion which is a big part of this paper, giving a personal interpretation of the results.

3-I cannot see table 1 (and details) and scheme 1 (and details) in the discussion, but I suppose they have some utility…

4-in 3.1 and 3.2.2 and 3.2.3 and 3.2.4 and 3.2.5, there are % but not in figure 2 and in figure 1 (there are ratios with a scale from 0 to 1). Moreover precise in the caption of these figures the meaning of "3" "4" 6" and "7", for a rapid reading.

 5-some sentences of the authors' answer in the paragraph "the reviewer feels..." should be put at the beginning of the discussion or at the beginning of 3.2, they are much more clear than in the present revision of the paper (especially the "indirect" study), and will allow a non-specialist to understand this interesting paper, especially for a special issue devoted to Plant Aphid Interactions.

 6-As minor point, in the discussion, there is a (Error! Reference source not found.) [50].

Author Response

The response can be found in the letter to the academic editor.